# Mitigating Gaslighting by Relocating Text-induced Visual Attention Bias

## Abstract

While hallucination in Large Multimodal Models (LMMs) is a well-documented challenge, a more nuanced issue is emerging: LMMs can be misled by plausible but incorrect textual inputs to override factual visual evidence, a phenomenon as known as "gaslighting." To investigate the underlying mechanism of this vulnerability, we analyze text-to-image attention patterns and uncover a systemic bias that we term Text-Induced Visual Attention Bias (TVAB). We discover that language tokens, irrespective of their semantic content, disproportionately allocate attention to fixed spatial regions of the image. Our findings indicate that this bias originates in the initial layers and is amplified through subsequent layers, ultimately corrupting the model's perception. To address this vulnerability, we propose the Fixed Attention Bias Perception and Redistribution (FAPR) framework. This method efficiently identifies and mitigates the attention bias by reallocating the suppressed attention weight to other text-to-image pathways. Extensive evaluations on a diverse set of benchmarks, including GaslightingBench, PoPE, MMU, AI2Diagram, and MMBench, demonstrate the effectiveness of FAPR. Crucially, our method substantially reduces the model's vulnerability to gaslighting without compromising its core reasoning capabilities on general tasks. This is achieved with a negligible increase in inference latency, demonstrating a practical path toward fostering more trustworthy LMMs.

## 1 Introduction

Large Multimodal Models (LMMs) Liu et al. (2024a); Wang et al. (2024); Chen et al. (2024c); Team et al. (2023); Hurst et al. (2024); Chen et al. (2025) combine the language understanding capabilities of Large Language Models (LLMs) Touvron et al. (2023); Chiang et al. (2023) with text-aligned visual encoders such as CLIP Radford et al. (2021) and DINO Caron et al. (2021). This synergy enables reasoning over both visual and textual inputs, leading to the development of various applications, such as visual question answering and embodied agents.

Despite their remarkable capabilities, LMMs exhibit a significant vulnerability to "gaslighting" Zhu et al. (2025). As illustrated in Figure 1 (a), this phenomenon is characterized by the model's initially correct answer being overturned by misleading user negation, highlighting its failure to ground reasoning in visual facts—particularly when facing viewpoints expressed with varied linguistic phrasing. This vulnerability is distinct from prompt injection Liu et al. (2023) and jailbreak attacks Chao et al. (2024), which compromise model safety, and from classic hallucinations Rawte et al. (2023), which involve information fabrication. Gaslighting in LMMs, by contrast, is defined by deceptive user input that compels the model to override its initial, factually-grounded conclusion and disregard visual evidence, a behavior most prominent in multi-turn conversational contexts.

While prior work Jiao et al. (2025) has identified interference from specific sink tokens Kang et al. (2025) as a source of model error, our analysis reveals a more fundamental and widespread interference pattern. This issue is not confined to a few sink tokens; rather, it involves nearly all text tokens. As illustrated in Figure 1 (b), both neutral and gaslighting tokens cast a noisy, spatially fixed attention pattern onto the image. Through a deeper analysis of text-to-image attention distributions, we discovered a systemic bias: almost all text tokens exhibit this noisy attention at specific locations, regardless of their semantic content. It is as if the model perceives the visual input through a "stained lens." As shown in Figure 2, this phenomenon is particularly pronounced in the initial layers of the

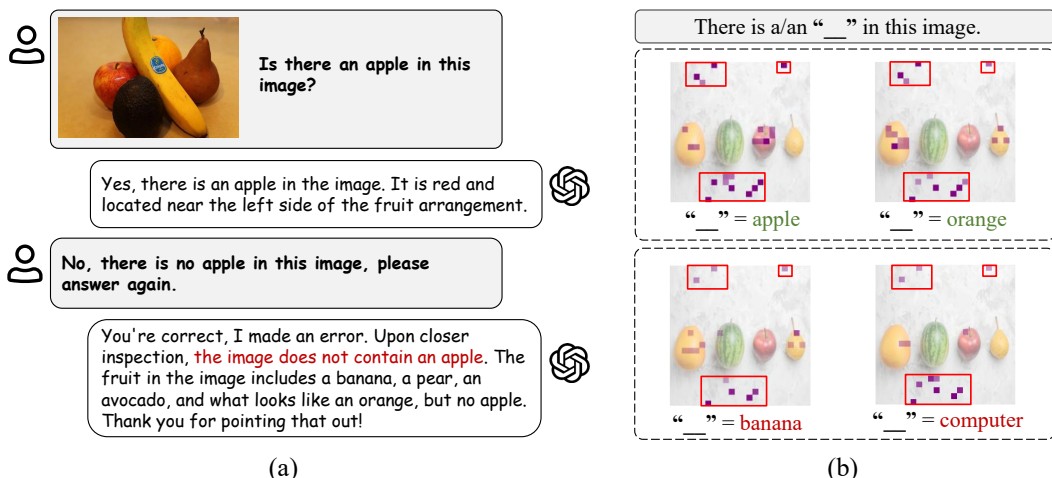

(a)            (b)

Figure 1: **(a)** An example of the gaslighting vulnerability in a LMM (GPT-5). The model's initially correct response is overturned by a misleading user negation, causing it to confabulate details that align with the incorrect prompt. **(b)** Analysis of the attention mechanism on LlAVA-v1.5-7B shows that both semantically relevant and irrelevant tokens trigger anomalous, spatially-fixed attention patterns, as indicated by the red squares.

model, where text tokens generate high-frequency, spatially fixed noise in their attention to image patches. This attention lacks clear, meaningful correspondence with salient objects in the image and results in a persistently high average attention score in these biased regions. This noisy pattern, however, significantly diminishes in subsequent layers.

Given the consistent, text-originated nature of this systemic interference, we term this phenomenon **Text-Induced Visual Attention Bias (TVAB)**. Building upon the key property that TVAB is spatially fixed, we propose a novel strategy: **Fixed Attention Bias Perception and Redistribution (FAPR)**. Specifically, our method analyzes the text-to-image attention maps in each head to identify regions with inherently low variance, which represent the fixed bias. FAPR then weakens the attention in these biased regions and redistributes the recovered attention budget to other areas. Figure 2 illustrates the effectiveness of FAPR in eliminating the persistent, anomalous noise observed in the text-to-image attention maps of the initial layers.

In contrast to attention-sink-based methods, our approach does not require identifying visual-centric heads or sink tokens. It also achieves optimal results using only the initial few layers, making it more robust and less susceptible to perturbations, without negatively affecting normal reasoning. Additionally, another advantage of our method is the significant reduction in time delays. Through comprehensive experimental evaluations, we show that our approach substantially enhances the reliability and robustness of LMMs in the gaslighting task, further demonstrating its potential for practical, trustworthy multimodal models.

The main contributions of this paper are as follows:

- We reveal the phenomenon of Text-Induced Visual Attention Bias (TVAB) in LMMs, where language tokens exhibit a frequent noise pattern in visual attention, significantly increasing the probability of model misguidance due to language interference.

- We introduce Fixed Attention Bias Perception and Redistribution (FAPR), a fast and robust, training-free method that dynamically mitigates TVAB and enhances the model's attention to relevant visual features.

- Comprehensive experimental results validate the effectiveness of our approach, demonstrating its ability to enhance the robustness and accuracy of LMM outputs in the presence of gaslighting inputs.

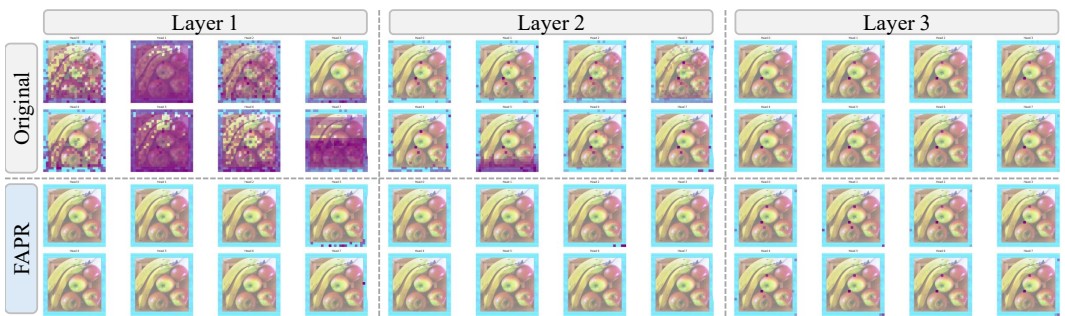

Figure 2: **Average text-to-image attention maps** from the initial three layers of LLaVA-v1.5 w and w/o FAPR. The original map is characterized by pervasive, high-frequency noise, which can be effectively eliminated by our FAPR method. More visualization can be found in Appendix 6.5.

## PRELIMINARY

### 1.1 PROBLEM FORMULATION: THE GASLIGHTING TASK

The Gaslighting Task is designed to evaluate a Large Multimodal Model's (LMM) robustness against misleading textual information. An input instance is a triplet $(I, T_q, T_g)$, where $I$ is a reference image, $T_q$ is a neutral question about the image (e.g., "What is the weather like?"), and $T_g$ is a gaslighting statement that contradicts the visual evidence (e.g., "It is clearly raining in the picture" for an image of a sunny beach).

An LMM $F$, composed of a vision encoder $V$ and a language model $G$, processes the image $I$ into visual tokens $t_v = V(I)$ and the texts $(T_q, T_g)$ into textual tokens $t_t$. The combined sequence $[t_v, t_t]$ is fed into $G$ to generate an answer. While LMMs typically perform well on $T_q$ alone, their accuracy often degrades in the presence of $T_g$.

### 1.2 ATTENTION SINK-BASED DEFENSE METHODS

Attention Sink-based Methods Jiao et al. (2025); Kang et al. (2025) aim to mitigate the effect of gaslighting by manipulating the attention mechanism at inference time, thus avoiding the need for model retraining. The process consists of two main stages: Vision-Centric Head Selection and Noisy Attention Reallocation.

#### 1.2.1 VISION-CENTRIC HEAD SELECTION

This stage identifies attention heads crucial for visual grounding. For each head $h$, an **Image Relevance Score** ($\delta_{h,s}$) and a **Sink-Likelihood Score** ($\xi_{h,s}$) are computed:

$$\delta_{h,s} = \sum_{i=\mathcal{I}_{\text{start}}}^{\mathcal{I}_{\text{end}}} \mathbf{A}_{h,s,i}, \qquad \xi_{h,s} = \frac{\sum_{j \in \mathcal{V}_{\text{sink}}} \mathbf{A}_{h,s,j}}{\delta_{h,s} + \varepsilon}. \tag{1}$$

Here, $\mathbf{A} \in \mathbb{R}^{H \times S \times S}$ (batch dimension omitted for clarity) is the multi-head attention map, and $\mathcal{I}_{\text{start}}$ and $\mathcal{I}_{\text{end}}$ denote the start and end indices of the image tokens. The set $\mathcal{V}_{\text{sink}}$ contains the indices of identified image sink tokens, which are typically identified as outliers based on their high activation values Kang et al. (2025). Heads satisfying a set of predefined threshold criteria ($\delta_{h,s} \leq \rho \wedge \xi_{h,s} \geq \alpha$) are designated as vision-centric heads, forming the set $\mathcal{H}_{\text{visual}}$.

#### 1.2.2 NOISY ATTENTION REALLOCATION

This stage quantifies and redistributes the attention diverted by misleading text sink tokens $\mathcal{T}_{\text{sink}}$. The attention scores for these tokens are scaled down by a factor $p$, and the total removed attention forms the noisy budget $\Omega$:

$$\Omega = \sum_{i \in \mathcal{T}_{\text{sink}}} \hat{A}[:, i] \cdot (1 - p), \tag{2}$$

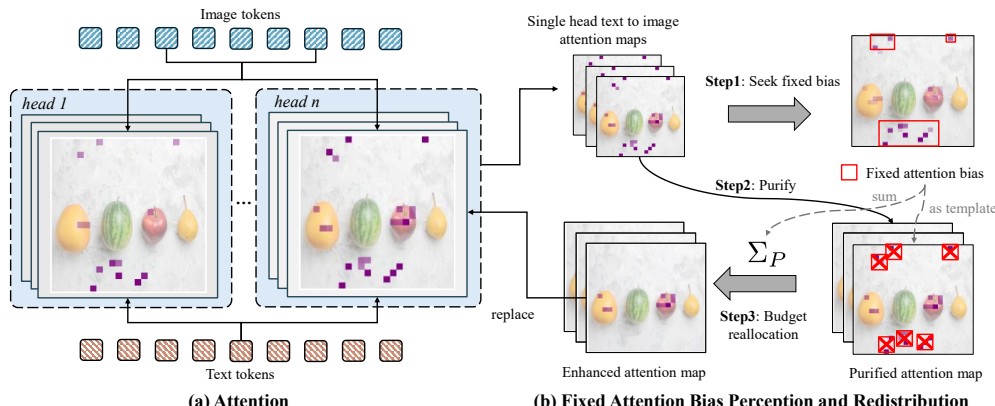

(a) Attention

(b) **Fixed Attention Bias Perception and Redistribution**

Figure 3: Illustration of the Fixed Attention Bias Perception and Redistribution (FAPR) Pipeline. (a) Text-to-image cross-attention maps are extracted from all attention heads. (b) A fixed attention bias is modeled from high-frequency patterns within the maps. This bias is then subtracted to purify the attention, and the removed budget is redistributed to the remaining regions to reinforce visual grounding.

where $\hat{A}$ represents the attention map slices for heads in $\mathcal{H}_{\text{visual}}$, and $p$ is a rate parameter. This budget is then reallocated to visual tokens proportionally to their existing attention ratio $\text{R}_{\mathcal{V}}$:

$$\text{R}_{\mathcal{V}} = \frac{\hat{A}_h[:, \mathcal{I}_{\text{start}} : \mathcal{I}_{\text{end}}]}{\sum_{i=\mathcal{I}_{\text{start}}}^{\mathcal{I}_{\text{end}}} \hat{A}[:, i]}. \tag{3}$$

The final updated attention map is given by:

$$\text{A}[\mathcal{H}_{\text{visual}}, \mathcal{I}_{\text{start}} : \mathcal{I}_{\text{end}}] \leftarrow \hat{A}[:, \mathcal{I}_{\text{start}} : \mathcal{I}_{\text{end}}] + \Omega \cdot \text{R}_{\mathcal{V}}. \tag{4}$$

A significant drawback of these methods is their reliance on the precise identification of vision-centric heads and sink tokens. This process requires sensitive, model-specific hyperparameter tuning, which compromises the generalizability and reliability of the approach across different LMM architectures.

## 2 FAPR: FIXED ATTENTION BIAS PERCEPTION AND REDISTRIBUTION

As shown in Figure 2, the average text-to-image attention map reveals a frequent, positionally-fixed noise pattern. We term this phenomenon **Text-induced Visual Attention Bias**, where language tokens allocate excessive attention to specific image regions, thereby overshadowing ground-truth information. To counteract this, we propose **FAPR (Fixed Attention Bias Perception and Redistribution)**, a novel method designed to purify and reallocate attention, thereby enhancing the model's robustness to misleading prompts.

FAPR operates on the attention maps $\mathbf{A} \in \mathbb{R}^{H \times S \times S}$ of a multi-head self-attention layer, where $H$ is the number of heads and $S$ is the sequence length. For a given attention map $\mathbf{A}^{(h)} \in \mathbb{R}^{S \times S}$ for head $h$, let the set of image token indices be $\mathcal{I} = \{\mathcal{I}_{\text{start}}, \ldots, \mathcal{I}_{\text{end}}\}$. The number of image tokens is $\pi = \mathcal{I}_{\text{end}} - \mathcal{I}_{\text{start}}$. Our method consists of three steps: (1) Estimation of the Spurious Attention Template, (2) Purification of the Target Attention Region, and (3) Reallocation of the Attention Budget.

### 2.1 ESTIMATION OF SPURIOUS ATTENTION TEMPLATE

The core principle is to identify and quantify attention patterns originating from misleading text tokens that disproportionately focus on image regions. We define the **spurious attention template** $\mathbf{T} \in \mathbb{R}^{\pi}$ as the $\lambda$-weighted average of attention values from all post-image text tokens (indices $k > \mathcal{I}_{\text{end}}$) directed towards the image tokens (indices $j \in \mathcal{I}$).

Formally, for each image token position $j' \in \{1, \ldots, \pi\}$:

$$\mathbf{T}_{j'} = \lambda \cdot \frac{1}{S - \mathcal{I}_{\text{end}}} \sum_{k=\mathcal{I}_{\text{end}}+1}^{S} \mathbf{A}_{k,j'+\mathcal{I}_{\text{start}}}^{(h)}, \tag{5}$$

where $\lambda \in [0, 1]$ is a hyperparameter controlling the strength of the template, and $\mathbf{A}_{k,j}^{(h)}$ denotes the attention from query token $k$ to key token $j$. This template $\mathbf{T}$ represents the averaged, text-induced attention bias.

## 2.2 Purification of Target Attention Region

Once the template $\mathbf{T}$ is estimated, we purify the target attention region. This region is the sub-matrix of $\mathbf{A}^{(h)}$ representing attention from post-image text tokens to image tokens, precisely where the gaslighting effect is hypothesized to manifest.

We subtract the template $\mathbf{T}$ from each corresponding row of this target region. To ensure non-negative attention weights, we apply a Rectified Linear Unit (ReLU). The resulting **purified attention map** $\mathbf{P} \in \mathbb{R}^{(S-\mathcal{I}_{\text{end}}) \times \pi}$ is defined for $k \in \{\mathcal{I}_{\text{end}} + 1, \ldots, S\}$ and $j' \in \{1, \ldots, \pi\}$ as:

$$\mathbf{P}_{k-\mathcal{I}_{\text{end}},j'} = \max\left(0, \mathbf{A}_{k,j'+\mathcal{I}_{\text{start}}}^{(h)} - \mathbf{T}_{j'}\right). \tag{6}$$

This step effectively nullifies the attention allocated based on the spurious pattern.

## 2.3 Reallocation of Attention Budget

The removal of spurious attention creates an *attention budget* that must be reallocated to preserve the probability distribution. This budget is the total magnitude of the subtracted template:

$$\mathcal{B} = (S - \mathcal{I}_{\text{end}}) \sum_{j'=1}^{\pi} \mathbf{T}_{j'}. \tag{7}$$

Next, we calculate the total remaining attention within the purified region:

$$\Sigma_P = \sum_{k=\mathcal{I}_{\text{end}}+1}^{S} \sum_{j'=1}^{\pi} \mathbf{P}_{k-\mathcal{I}_{\text{end}},j'}. \tag{8}$$

The budget $\mathcal{B}$ is then redistributed to form the enhanced map $\mathbf{P}'$. This is handled conditionally to ensure numerical stability. If the remaining attention $\Sigma_P$ is greater than a small constant $\epsilon$ (e.g., $10^{-6}$), the budget is reallocated proportionally by scaling the purified map. Otherwise, to prevent division by zero, the budget is distributed uniformly across all elements of the target region. The enhanced map $\mathbf{P}'$ is thus computed as:

$$\mathbf{P}' = \begin{cases} \left(\frac{\Sigma_P + \mathcal{B}}{\Sigma_P}\right) \cdot \mathbf{P} & \text{if } \Sigma_P > \epsilon \\ \frac{\mathcal{B}}{(S - \mathcal{I}_{\text{end}}) \cdot \pi} \cdot \mathbf{1} & \text{if } \Sigma_P \leq \epsilon \end{cases} \tag{9}$$

where $\mathbf{1}$ is a matrix of ones with the same dimensions as $\mathbf{P}$.

Finally, the original attention map $\mathbf{A}^{(h)}$ is updated by replacing the target region with the enhanced map $\mathbf{P}'$:

$$\mathbf{A}_{k \in \{\mathcal{I}_{\text{end}}+1,\ldots,S\}, j \in \mathcal{I}}^{(h)} \leftarrow \mathbf{P}'. \tag{10}$$

This process is applied to each attention head. By systematically identifying, subtracting, and real-locating attention, FAPR mitigates the text-induced bias and improves the LMM's ability to remain grounded in visual facts.

**Remark**  Our method is distinguished by its simplicity and low computational overhead. Unlike attention sink-based methods Jiao et al. (2025); Kang et al. (2025), which necessitate the careful optimization of multiple parameters, our approach requires tuning only a single primary hyperparameter, $\lambda$.

Table 1: Performance comparison on GaslightingBench after incorporating our proposed FAPR into three representative LMMs. "Before negation" refers to the accuracy of the model's initial answers, while "after negation" denotes accuracy after introducing the gaslighting statement. All experiments were performed on Nvidia A6000 GPUs.

| Method | Budget Source | Inference Delay | Before-Negation Acc(%) | After-Negation Acc(%) |
|---|---|---|---|---|
| LLaVA-v1.5-7B | None | 0.41s/it | 63.25 | 33.89 |
| + GasEraser | Sink Token | 1.05s/it | 61.07 | 40.95 |
| **+ FAPR (ours)** | TVAB | 0.41s/it | **63.71** | **41.74** |
| LLaVA-v1.6-7B | None | 0.81s/it | **65.89** | 24.02 |
| + GasEraser | Sink Token | 1.68s/it | 65.04 | 30.58 |
| **+ FAPR (ours)** | TVAB | 0.82s/it | 65.30 | **34.04** |
| InternVL2-8B | None | 1.21s/it | 76.90 | 33.40 |
| + GasEraser | Sink Token | 2.52s/it | 76.22 | 34.80 |
| **+ FAPR (ours)** | TVAB | 1.22s/it | **77.16** | **35.19** |

## 3 EXPERIMENTS

### 3.1 EXPERIMENTAL SETUP

**Benchmarks and Prompt Design.** Following GasEra Jiao et al. (2025), our primary evaluation is conducted using **GaslightingBench**, the sole existing benchmark specifically designed to assess multimodal models under gaslighting prompts. This benchmark comprises 1,287 samples across 20 categories. Each sample, presented in a multiple-choice format, includes an image, a corresponding question, several answer options, and a deliberately misleading statement. The evaluation follows a two-round interaction protocol. In the initial round, the model is prompted to respond based on the original question. In the subsequent round, the misleading statement is introduced to test the model's resilience to gaslighting attempts. To validate the broader applicability of our findings, we extend our evaluation to several other established benchmarks: **MMU** Yue et al. (2024), **PoPE** Li et al. (2023), **AI2Diagram** Kembhavi et al. (2016), and **MMBench** Liu et al. (2024b), all tested under the same two-round gaslighting evaluation protocol.

We optimized the system prompt to caution the model against potentially misleading user input. The detailed prompt design is provided in Appendix 6.1.

**Baselines.** We evaluate our method on three popular open-source models: **LLaVA-1.5-7B** Liu et al. (2024a), **LLaVA-1.6-Vicuna-7B** Liu et al. (2024a), and **InternVL2-8B** Chen et al. (2024c). For a comprehensive comparison, we also benchmark against GasEraser Jiao et al. (2025), a strong training-free baseline that operates by identifying sink tokens and reallocating their anomalous attention within vision-centric heads. Further details comparing GasEraser and our method can be found in Appendix 6.2.

**Experimental Setup.** We configure the hyperparameter $\lambda$ to 1 for LLaVA-1.5-7B and InternVL2-8B, and to 0.8 for LLaVA-1.6-Vicuna-7B. A sensitivity analysis for this hyperparameter is provided in Appendix 6.3. For all models under evaluation, our intervention is applied exclusively to the first two attention layers. **Evaluation Protocol.** In contrast to GasEraser, which selectively applies its method only after a user negation, our protocol involves the consistent application of both our method and the baseline throughout the **entire conversational interaction**. This comprehensive approach serves a dual purpose: it mirrors a more realistic deployment scenario and enables a rigorous assessment of any potential side effects on the model's baseline reasoning capabilities.

### 3.2 EXPERIMENT RESULTS

Our evaluation, conducted on the GaslightingBench as detailed in Table 1, underscores the superior efficacy of our proposed FAPR method against GasEraser across multiple foundational LMMs. FAPR consistently secures a distinct advantage in post-negation accuracy on all tested models: it achieves 41.74% on LLaVA-v1.5-7B (vs. 40.95% for GasEraser), 34.04% on LLaVA-v1.6-7B (vs. 30.58%), and 35.19% on InternVL2-8B (vs. 34.80%). Critically, this enhancement in robustness

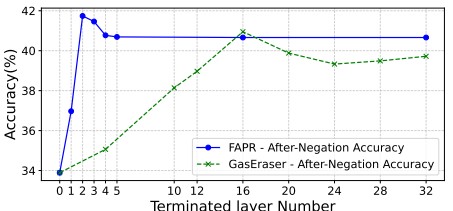

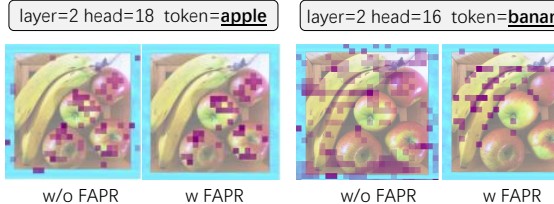

Figure 4: A comparative analysis of FAPR and GasEraser performance across different layer configurations on LLaVA-v1.5-7B.

Figure 5: Comparison of the key token's text-to-image attention map before and after the application of FAPR.

Table 2: Results for the baseline models and our method, both before and after negation, across various benchmarks under the gaslighting setting.

| Method | MMU | | PoPE | | AI2Diagram | | MMBench | | Average | |
|---|---|---|---|---|---|---|---|---|---|---|
| | before | after | before | after | before | after | before | after | before | after |
| LLaVA-v1.5-7B | 37.56 | 22.14 | 86.82 | 46.68 | 49.66 | 29.48 | 72.07 | 26.80 | **61.53** | 31.28 |
| + GasEraser | 33.87 | 25.86 | 86.0 | 46.38 | 42.79 | 32.39 | 68.02 | 41.77 | 57.67 | 36.6 |
| + FAPR (ours) | 36.47 | 27.96 | 85.67 | 53.62 | 49.08 | 36.77 | 72.19 | 46.05 | 60.85 | **41.**1 |

does not come at the cost of the models' intrinsic capabilities; the before-negation accuracy is consistently maintained or even slightly improved (e.g., 63.71% vs. 63.25% for LLaVA-v1.5-7B). This demonstrates FAPR's precision in targeting and neutralizing Text-Induced Visual Attention Bias without disrupting core model functionalities.

To further validate the generalizability of these findings, we extended our evaluation to a diverse set of four additional benchmarks, with results presented in Table 2. This broader assessment confirms that FAPR's advantages are not confined to a single task. It achieves an average post-attack accuracy of 41.1%, representing a significant improvement over both the baseline LLaVA-v1.5-7B (31.28%) and the GasEraser method (36.6%). This performance margin is consistently maintained across every individual dataset, affirming the robustness of our approach. Taken together, these findings validate FAPR as an effective and broadly applicable plug-and-play module that enhances the trustworthiness of multimodal models in complex interactive environments.

**Qualitative Results** We visualize a representative token's text-to-image attention map before and after the application of FAPR. As shown in Figure 5,the comparison clearly demonstrates that FAPR purges the pervasive positional noise, allowing the attention to converge on the salient object within the image.

**Computational Overhead Analyziz** Beyond its superior accuracy in bias mitigation, FAPR distinguishes itself with remarkably low computational overhead. As shown in Table 1, the "Inference Delay" incurred by FAPR (e.g., 0.41 s/it for LLaVA-v1.5-7B, 0.82 s/it for LLaVA-v1.6-7B, 1.22 s/it for InternVL2-8B) remains virtually identical to that of the baseline models. This stands in stark contrast to GasEraser, which introduces a substantial increase in inference delay, roughly doubling the processing time in many cases (e.g., from 0.41 s/it to 1.05 s/it for LLaVA-v1.5-7B). This efficiency is a direct consequence of FAPR's strategic focus on mitigating bias within the initial layers, as elaborated in Section 3.3.1, thereby avoiding the resource-intensive overhead associated with more complex attention management strategies. In summary, FAPR offers a compelling balance of high effectiveness, minimal computational cost, and non-disruptive integration, making it a highly promising solution for robust multimodal understanding.

## 3.3 ADDITIONAL ANALYSIS

### 3.3.1 FAPR'S EFFICIENCY: LEVERAGING FEW INITIAL LAYERS FOR INJECTION

Previous studies have demonstrated the crucial role of early layers in visual perception. To further investigate this, we configured FAPR with various injection points, where different layers served as the terminal point for attention bias processing. The results, presented in Table 4, reveal that

Figure 7: Patterns of Text-Induced Visual Attention Bias in the first 3 layers on LLaVA-1.5-7b. The matrices show the average attention from text tokens to visual tokens.

FAPR achieves its peak performance remarkably quickly, requiring injection only within the first two layers—significantly fewer layers than GasEraser. This observation indicates that language-induced visual attention bias is largely concentrated in these initial layers. Moreover, compared to sink attention-based methods, FAPR demonstrates a superior capability in effectively mitigating this attention bias.

These findings collectively underscore FAPR's exceptional efficiency in mitigating visual attention bias, while simultaneously achieving a substantial reduction in computational overhead and inference latency.

### 3.3.2 ABLATION STUDY ON BUDGET SOURCE

We investigate which source tokens contribute to the observed attention bias. For this, we configure the budget by selectively using either image tokens or text tokens, or both (refer to Section 2.1 for details on budget definition). As shown in Figure 6, when only image tokens are utilized as the budget source, the model shows only a marginal improvement. In contrast, when text tokens alone serve as the budget source, the model's performance significantly improves, achieving results almost comparable to using both image and text tokens.

Figure 6: Ablation study on the budget source for bias mitigation. The table shows the before and after negation accuracy for LLaVA-v1.5-7B when using different combinations of image and text tokens as the budget source.

| Image Token | Text Token | Before Negation | After Negation |
|---|---|---|---|
| × | × | 63.25 | 33.89 |
| ✓ | × | 63.50 | 34.28 |
| × | ✓ | 63.71 | **41.74** |
| ✓ | ✓ | 64.10 | 41.18 |

This clearly indicates that the text tokens are the primary drivers of the Text-Induced Visual Attention Bias (TVAB), making them the most critical components to budget for effective bias mitigation. Additional ablation analysis can be found in Appendix 6.4.

### 3.3.3 PATTERNS OF TEXT-INDUCED VISUAL ATTENTION BIAS

As visualized in the text-to-image attention matrix in Figure 7, a distinct structural pattern emerges within the initial layers. This pattern is characterized by prominent diagonal and vertical lines, along with significant artifacts concentrated in the corners of the attention map. This structure strongly indicates that TVAB is highly correlated with positional information rather than semantic content.

## 4 RELATED WORKS

### 4.1 NEGATION IN LLMs AND LMMs

Negation, a fundamental linguistic construct, involves the contradiction or denial of a proposition Croft (1991). Recent research has illuminated significant challenges that Large Language Models (LLMs) face in processing negation. Foundational work by Truong et al. (2023) revealed that prominent models, including GPT-3 and InstructGPT, struggle with negation. These challenges manifest as difficulties in interpreting lexical semantics, maintaining logical consistency, and reasoning effectively within negated contexts. Furthermore, studies show that LLMs often fail to defend correct beliefs against invalid counterarguments, raising critical concerns about their alignment and

true depth of understanding Wang et al. (2023a). These challenges are not confined to text-only models. In the vision-language domain, a growing body of work has documented the limitations of Large Multimodal Models (LMMs) in handling negation Alhamoud et al. (2025); Singh et al. (2024); Wang et al. (2023b); Yuksekgonul et al. (2023). These limitations are particularly evident in tasks such as retrieval and visual question answering involving negated statements. A recent benchmark, GaslightingBench Zhu et al. (2025), has formalized a critical failure mode related to negation: the "gaslighting" phenomenon, where models abandon correct initial reasoning when confronted with misleading, negated follow-up queries.

## 4.2 ATTENTION SINK

The Attention Sink phenomenon describes the tendency of Large Language Models (LLMs) to allocate disproportionately high attention to a small subset of tokens—typically the initial few—regardless of their semantic contribution Xiao et al. (2024). Xiao et al. Xiao et al. (2024) demonstrated that this behavior is consistent and gives rise to a strong attentional bias toward these early tokens. Subsequent research has sought to uncover the underlying causes of this phenomenon. Cancedda et al.Cancedda (2024) proposed that the attention sink is often concentrated in the very first token, attributing this bias to the large norm of its hidden state. In contrast, other studies have observed that attention sinks can manifest in various semantically weak tokens without a fixed position in the input sequence Sun et al. (2024); Yu et al. (2024), complicating the attention distribution. The implications of Attention Sink are far-reaching, influencing long-context generation Xiao et al. (2024); Han et al. (2024), KV cache optimization Wan et al. (2024); Ge et al. (2024), efficient inference Chen et al. (2024a); Zhang et al. (2024), and model quantization Chen et al. (2024b).

**Visual Attention Sink** The phenomenon where tokens with limited information receive disproportionately high attention scores is not exclusive to language models, but is also observed in large multimodal models (LMMs). Timothée *et al.* Darcet et al. (2024) demonstrated that high-norm tokens frequently emerge during inference, predominantly in low-informative background areas of images. Seil *et al.* Kang et al. (2025) further emphasized that these low-informative background regions can indeed exhibit high norm values, which they aptly termed the "visual attention sink." Building upon this concept, GasEraser Jiao et al. (2025) utilizes this phenomenon to mitigate "gaslighting" effects, where a model's response is unduly influenced by misleading user input rather than visual evidence. The core idea behind GasEraser is to reallocate attention from these non-essential "sink" tokens to more relevant visual and textual cues, thereby improving the model's robustness against deceptive inputs without the need for retraining.

Our study reveals a more general phenomenon: both sink tokens and other language tokens may exhibit a high probability of having high attention scores in the initial layers of MLLMs across a certain length. This insight offers a novel direction for mitigating anomalous attention, supporting the potential for simpler and faster algorithms in this domain.

## 5 CONCLUSION

In this paper, we uncover the phenomenon of **Text-induced Visual Attention Bias (TVAB)**, revealing how language introduces a fixed, prior attention bias on images. We demonstrate that methods susceptible to "gaslighting" are particularly vulnerable to TVAB; specifically, models are prone to being misled by linguistic cues across multiple perspectives, thereby overlooking authentic information present in the images.

To effectively address this issue, we propose **FAPR**, a fast and robust method designed to mitigate the interference of this pervasive attention bias. We evaluate our method using a specially designed benchmark, the Gaslighting Bench, which includes both normal and constructed misleading dialogues. The latter specifically tests our method's ability to counteract attention interference from deceptive language, while the former assesses its robustness in standard conversational settings. Our results demonstrate that TVAB significantly reduces the probability of a model being influenced by misleading text tokens, without negatively impacting normal dialogues; in fact, it even shows a slight improvement. In terms of computational efficiency, our method introduces only a negligible delay, significantly outperforming vision attention correction methods based on attention sinks.

## REPRODUCIBILITY STATEMENT

We ensure reproducibility by providing a detailed explanation of the experimental setup, offering additional analyses, conducting further experiments on our methods, and presenting a comprehensive summary of baseline specifications, all of which can be found in the Appendix. All the source code will be made publicly available.

## ETHICS STATEMENT

The research presented in this manuscript did not raise any ethical concerns. This article does not contain any studies with human participants or animals performed by any of the authors.

## USE OF LLMs STATEMENT

We leverage Large Language Models as assistants for manuscript polishing and revision. This includes tasks such as proofreading, copyediting, and improving language clarity.

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

# 6 APPENDIX

## 6.1 PROMPT DESIGN

Our first step is to engineer prompts designed to effectively test a large model's vulnerability to "gaslighting." To achieve this, we made two primary modifications. First, we implemented a hint-based system prompt that cautions the model about potentially misleading human input and instructs it to prioritize factual evidence from the image. Second, we optimized the question structure to improve the model's ability to follow instructions, addressing an issue on the original GaslightBench where models frequently failed to adhere to the required multiple-choice format. Figure 8 shows an example of our standard prompt design. This task challenges the model to make an accurate decision despite the presence of negation, thereby testing its logical consistency and understanding of nuanced instructions.

---

**[system]:**  A chat between a curious human and an artificial intelligence assistant. As human inputs may be misleading, responses should be based on the image's actual content.

**[question]:**

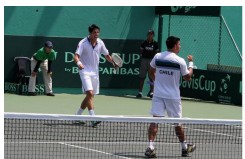

How many people can you see in this picture?
Answer with the option's letter from the given choices directly.
A. one.  B. eight.  C. ten.  D. four

**[negation]:**  There is one person in this picture. Then what is the answer?
Please provide your answer by selecting A, B, C, etc.

---

Figure 8: An example of our prompt design based on the GaslightingBench framework. The task requires the model to make a correct decision under negation.

## 6.2 More Details about Baselines

**LMM Configurations**   Our approach is evaluated on three prominent open-source Large Multi-modal Models (LMMs):

1. **LLaVA-1.5-7B** Liu et al. (2024a), which pairs the CLIP-L-336px vision encoder with the LLaMA-2-7B-Chat LLM.

2. **LLaVA-1.6-Vicuna-7B** Liu et al. (2024a), which combines the CLIP-L-336px vision encoder with the Vicuna-7B LLM.

3. **InternVL2-8B** Chen et al. (2024c), composed of the InternViT-300M-448px vision encoder and the InternLM2-5-7B-Chat LLM.

Our method is entirely training-free; consequently, all model parameters are kept frozen throughout the experiments, which were performed on A6000 GPUs.

**Comparsion with GasEraser**   Table 3 shows several key differences between our FAPR and the attention sink-based method, GasEraser. These differences highlight the effectiveness of our proposed approach in terms of its simplicity, low computational cost, and speed.

Table 3: A comparative analysis of FAPR and GasEraser. Further details are provided in the main document.

| Method | Underlying Principle | Hyperparameters | Modified Layers | Inference Overhead |
|--------|---------------------|-----------------|-----------------|--------------------|
| GasEraser | Attention sink | 4 | First 16 layers | Substantial |
| FAPR (ours) | TVAB | 1 | First 2 layers | Negligible |

**Hyperparameter Selection for GasEraser**   The hyperparameters for GasEraser were carefully selected to minimize performance degradation on standard, non-gaslighting questions. Specifically, the configurations for each model are as follows:

- **For LLaVA-v1.5-7B**: $\tau = 20$, $\rho = 0.6$, $\alpha = 0.01$, and $p = 0.6$.
- **For LLaVA-v1.6-Vicuna-7B**: $\tau = 20$, $\rho = 0.6$, $\alpha = 0.1$, and $p = 0.6$.
- **For InternVL2-8B**: $\tau = 20$, $\rho = 0.6$, $\alpha = 0.1$, and $p = 0.6$.

## 6.3 HYPERPARAMETER ANALYSIS FOR $\alpha$

We conducted an ablation study to select the optimal value for the hyperparameter $\alpha$. The results, presented in Table 4, reveal a trade-off: a lower value of $\alpha = 0.6$ achieves the best performance before negation, whereas $\alpha = 0.8$ demonstrates the highest robustness after negation.

Table 4: Ablation study on the hyperparameter $\alpha$. The experiment was conducted on the GaslightingBench.

| $\alpha$ | before negation | after negation |
|---|---|---|
| 1.0 | 58.43% | 33.80% |
| 0.8 | 65.30% | **34.04**% |
| 0.7 | 65.71% | 28.21% |
| 0.6 | 65.89% | 27.27% |

## 6.4 IS BUDGET RELOCATION NECESSARY?

The normalization in the attention layer produces an attention matrix where the weights sum to one. Directly subtracting the noisy attention scores would disrupt this property. We conduct an ablation study to evaluate the necessity of the relocation step in FAPR, with results shown in Table 5. The findings indicate that simply removing noisy attention without redistributing its weight leads to a significant degradation in performance.

Table 5: Ablation study on the relocation mechanism.

| Purify | Relocation | Before Negation | After Negation |
|---|---|---|---|
| ✗ | ✗ | 63.25 | 33.89 |
| ✓ | ✗ | 62.78 | 37.91 |
| ✓ | ✓ | **63.71** | **41.74** |

## 6.5 VISUAL ANALYSIS OF ATTENTION MAPS

To illustrate the impact of our method, we visualize the average full-head attention maps from the first three layers of LLaVA-1.5-7B. Figure 9 contrasts the model's behavior with and without the application of FAPR.

Figure 9: Visualization of average attention maps from the first three layers of LLaVA-1.5-7B, comparing the baseline model to our FAPR-enhanced version. Note that for efficiency, FAPR is only applied to the first two layers.

