# OpenReview forum: "Mitigating Gaslighting by Relocating Text-induced Visual Attention Bias"
_ICLR.cc/2026/Conference — ICLR 2026 Conference Withdrawn Submission_

### Official Review · Reviewer_YiBw · 2025-10-17

**Soundness:** 2
**Presentation:** 1
**Contribution:** 2
**Rating:** 4
**Confidence:** 3

**Summary:**

This paper studies why large multimodal models (LMMs) can be “gaslighted” (flip a correct answer to a wrong one after a misleading statement) and attributes a core cause to a Text-Induced Visual Attention Bias (TVAB) that appears as a position-locked, high-frequency pattern in early layers. It proposes FAPR (Fixed Attention Bias Perception & Redistribution), an inference-time procedure that (i) estimates a spurious text-induced template over image tokens, (ii) subtracts it from text→image attention, and (iii) reallocates the removed probability mass. Across LLaVA-1.5-7B, LLaVA-1.6-7B, and InternVL2-8B on GaslightingBench, FAPR raises post-negation accuracy while keeping inference delay close to the base models; results are also reported under the same two-round protocol on MMU, PoPE, AI2Diagram, and MMBench.

# Motivation

* GaslightingBench evaluates models in two rounds: answer normally, then answer again after a misleading statement; the gap quantifies susceptibility to gaslighting.
* The authors introduce TVAB, a text-induced, spatially fixed attention pattern in early layers that can steer attention away from truly relevant regions.
* Prior “sink-token/head” defenses rely on selecting special tokens/heads and tuning several parameters; the paper targets a more general, position-fixed bias with a simpler, uniform treatment.

# Methodology

1. Estimate a spurious template $T$ over image positions by averaging attention from post-image text tokens to image tokens, scaled by a single hyperparameter λ (Eq. 5).
2. Purify the region by subtracting $T$ row-wise from the text→image submatrix and applying ReLU to keep weights non-negative, yielding $P$ (Eq. 6).
3. Reallocate the attention budget: compute the removed mass $B$ (Eq. 7) and remaining mass $\Sigma_P$ (Eq. 8); if $\Sigma_P>\epsilon$ scale $P$ to absorb $B$, else distribute $B$ uniformly, producing $P'$; replace the original submatrix with $P'$ (Eqs. 9–10).

# Evaluation Benchmarks

Benchmark GaslightingBench, two-round evaluation; also tested under the same protocol on MMU, POPE, AI2Diagram, and MMBench.

Latency observation. FAPR’s inference delay closely matches the base models; GasEraser roughly doubles delay in many cases.

# Analysis

* FAPR reaches peak gains by acting only in the first two layers, indicating the bias concentrates early; GasEraser needs deeper intervention.
* Visualizations show positional noise is purged and attention concentrates on salient objects after FAPR.
* The paper reports an ablation varying whether the reallocated “budget” comes from text tokens, image tokens, or both under the two-round protocol for LLaVA-1.5-7B.

**Strengths:**

* 1. Clear diagnosis of a failure mode (TVAB). The paper identifies a text-induced, position-locked bias in early text→image attention that amplifies through the network and correlates with gaslighting flips.
* 2. Simple, training-free defense (FAPR). The method is plug-and-play, acts on attention at inference, and uses one primary hyperparameter $λ$.
* 3. Minimal compute overhead. Empirically similar latency to the base models, unlike the sink-token baseline which roughly doubles delay in places.
* 4. Consistent robustness gains across models. Improves after-negation accuracy on LLaVA-1.5-7B, LLaVA-1.6-7B, and InternVL2-8B.
* 5. Narrow, principled intervention locus. Operating only in the first two layers aligns with the claim that TVAB originates early.

**Weaknesses:**

* 1. λ differs by model and application is fixed to the first two layers; general rules for selecting λ/layers across architectures and sizes aren’t demonstrated.
* 2. Robustness on open-ended generation (free-form VQA, captioning, CHAIR, LLaVA Bench(in-the-wild), .etc) under gaslighting isn’t reported.
* 3. ‘Before-negation’ accuracy is mostly preserved but shows minor downgrades, warranting further characterization of side effects.

**Questions:**

1. Do authors observe analogous bias templates in other sub-blocks (e.g., image→text attention)? If so, does extending FAPR there help or harm?
2. How sensitive are results to λ across datasets, prompt lengths, and image resolutions? Is there a default λ that works well across models, or these hyperparameters need to be adjusted for each of the new models?
3. Since the authors optimize the system prompt to warn about gaslighting, Ablation study with/without this prompt and report its independent effect, say, with different prompt, is recommended.
4. Will FAPR work on jailbreak/prompt-injection defenses, where there's obvious conflict between text and images as well? (positive or negative)?

---

> ### Author Response · Authors · 2025-11-20
>
> We appreciate your constructive comments. We address your main concerns below.
>
> **W1. Regarding Hyperparameters**
> Hyperparameters are inherent to existing methods addressing the visual attention sink. For example, [V] introduces four hyperparameters, whereas our method requires only one.
>
> **W2. Evaluation on Open-Ended Generation**
> We evaluated our method on the open-ended image captioning hallucination benchmark, CHAIR. The results, presented in the table below, demonstrate the effectiveness of our approach across different baselines.
>
> | Method | Time (s) | CHAIR$_i$ $\downarrow$ | Recall $\uparrow$ |
> | :--- | :--- | :--- | :--- |
> | **FAPR (Ours)** | **3.30** | 14.0 | 79.8 |
> | AGLA | 32.33 | 14.1 | 78.9 |
> | **AGLA + FAPR (ours)** | 32.37 | **13.5** | **80.14** |
> | ONLY | 3.32 | 14.3 | 75.7 |
> | **ONLY + FAPR (ours)** | 3.35 | **13.01** | **79.5** |
>
> **Reference:**
> [V] S. Kang, et al. Visual Attention Sink in Large Multimodal Models. ICLR 2025.

---

### Official Review · Reviewer_wE4A · 2025-10-29

**Soundness:** 3
**Presentation:** 3
**Contribution:** 3
**Rating:** 6
**Confidence:** 4

**Summary:**

This paper focuses on a newly identified vulnerability in large multimodal models—gaslighting, where misleading text overrides true visual evidence. The authors propose FAPR (Fixed Attention Bias Perception and Redistribution), a training-free inference-time method that identifies and suppresses biased attention regions and reallocates the attention budget to relevant visual features. Experiments across several LMMs (LLaVA 1.5/1.6, InternVL2) and multiple benchmarks (GaslightingBench, PoPE, MMU, AI2Diagram, MMBench) demonstrate improved robustness to gaslighting while maintaining normal reasoning and minimal inference delay.

**Strengths:**

1.The paper identifies and formalizes gaslighting as a novel vulnerability in large multimodal models, distinct from hallucination and prompt injection. It provides a new perspective for studying model trustworthiness.

2.The proposed training-free method, FAPR, efficiently suppresses text-induced visual attention bias and restores proper visual grounding.

3.The experiments, conducted on three major multimodal models and multiple benchmarks, show that FAPR effectively mitigates vulnerability to deceptive text while preserving reasoning ability and introducing almost no additional inference latency.

4.The paper is well-organized, readable, and technically detailed. The methodology is transparent and reproducible.

**Weaknesses:**

1.While the identification of TVAB is insightful, the paper would benefit from a more detailed theoretical discussion on the underlying mechanisms that lead to this bias.

2.The method is evaluated mainly on CLIP-based models; additional experiments on other architectures would strengthen the claim of general applicability.

3.Although the results are consistent across benchmarks, the overall performance gains are moderate, and further analysis of statistical significance could enhance credibility.

**Questions:**

1.How stable is the estimated spurious attention template (Eq. 5) across prompts or random seeds?

2.Did the authors attempt to visualize FAPR’s effect in later transformer blocks to confirm that redistribution effects persist?

3.Could FAPR interfere with multimodal tasks that require fine-grained visual grounding (e.g., referring expression segmentation)?

---

> ### Author Response · Authors · 2025-11-20
>
> We appreciate your constructive comments. We will provide a more comprehensive analysis of TVAB in the future submission, e.g., evaluations using different vision encoder backbones.

---

### Official Review · Reviewer_6ZDq · 2025-10-30

**Soundness:** 1
**Presentation:** 2
**Contribution:** 1
**Rating:** 2
**Confidence:** 5

**Summary:**

The paper addresses hallucination, by mitigating systematic attention bias induced by strong language priors. They propose Fixed Attention Bias Perception and Redistribution (FAPR), an attention reallocation approach that mitigates hallucination.

**Strengths:**

**Presentation**. The overall presentation is clear.

**Weaknesses:**

- **Novelty**. The paper overall follows AGLA [A], with marginal modifications. It should be pointed out that despite the very high relevancy to the mentioned paper, authors seem no discussion to clarify their difference to this work. Many aspects are alike, including,
    + Motivation. Figure 1 from this paper and Figure 2 from AGLA. Both discuss an experimental setup (change object query and see how attention patterns are affected. The conclusions are the same.)
    + Method. The authors propose an attention reallocation method, while AGLA also designs their approach in this way.

- **Comparison.** The paper only compares with GasEraser, while plenty of hallucination mitigation papers are ignored for comparison purpose.

[A] Mitigating Object Hallucinations in Large Vision-Language Models with Assembly of Global and Local Attention. CVPR 2025.

**Questions:**

No questions so far.

---

> ### Author Response · Authors · 2025-11-20
>
> Thank you for your feedback. We address your main concern below.
>
> **1. Generality of Attention Analysis**
> We respectfully clarify that text-token to image-token attention analysis of Figure 1 is not solely nor first used in AGLA [A]. Similar analysis methods have been employed in prior works, such as VAR [V]. Therefore, utilizing this analysis does not imply that our method follows the same technical trajectory as AGLA.
>
> **2. Key Differences from AGLA**
> There are substantial differences between AGLA and our proposed method in terms of setting, methodology, and efficiency:
>
> *   **Difference in Setting:** AGLA is designed to mitigate general **hallucination**, whereas our method specifically mitigates **gaslighting** by reducing the **Visual Attention Sink**.
> *   **Difference in Methodology:** AGLA leverages **complex contrastive decoding**, which incurs high computational costs. In contrast, our method uncovers the **Text-Visual Attention Bias (TVAB)** phenomenon and resolves it through a targeted, lightweight intervention.
> *  **Integration and Efficiency:**
> Our method is highly efficient and can be seamlessly integrated with existing hallucination mitigation frameworks. As results on CHAIR benchmark shown in the table below, AGLA suffers from significant latency (approx. 10x slower than ours). By combining our method with AGLA, we achieve the best performance with negligible additional cost to inference time.
>
> | Method | Time (s) | CHAIR$_i$ $\downarrow$ | Recall $\uparrow$ |
> | :--- | :--- | :--- | :--- |
> | **FAPR (ours)** | **3.30** | 14.0 | 79.8 |
> | AGLA | 32.33 | 14.1 | 78.9 |
> | **AGLA + FAPR (ours)** | 32.37 | **13.5** | **80.14** |
>
> [V] S Kang, et al. Visual Attention Sink in Large Multimodal Models. ICLR  2025.

---

### Official Review · Reviewer_gyzg · 2025-10-30

**Soundness:** 2
**Presentation:** 3
**Contribution:** 1
**Rating:** 2
**Confidence:** 5

**Summary:**

The paper presents a new method, Fixed Attention Bias Perception and Redistribution (FAPR), designed to mitigate the gaslighting vulnerability in Large Multimodal Models (LMMs), specifically addressing the issue of Text-Induced Visual Attention Bias (TVAB). While the paper introduces an interesting approach to the problem, there are several concerns that undermine its overall contribution.

**Strengths:**

The authors present a new method, Fixed Attention Bias Perception and Redistribution (FAPR), designed to mitigate the gaslighting vulnerability in Large Multimodal Models (LMMs).

**Weaknesses:**

1. The problem described by the authors is essentially a multi-turn dialogue problem, not a text-induced visual attention bias problem. In dialogue scenarios with only text, the user's input in the next turn can also lead the large model to give incorrect answers. According to the authors’ description, the scenario for the problem should involve both an image and misleading text input simultaneously, which would cause the model to exhibit attention bias toward the image. I believe the authors have not clearly defined the problem.
2. The authors only tested their method on two models, LLaVA-v1.5/1.6-7B and InternVL2-8B, and did not demonstrate the generalizability of their approach. Furthermore, it remains unclear whether the issue pointed out by the authors exists in larger-scale models.
3. Many multi-turn dialogue consistency enhancement methods could improve the issue mentioned by the authors. However, the authors did not include any comparison with these existing methods in the paper.
4. The case study presented in the paper is overly narrow, which makes me somewhat skeptical about the effectiveness of the proposed method.

**Questions:**

Please refer to the weakness.

---

> ### Author Response · Authors · 2025-11-20
>
> Thank you for your feedback. We address your main concern below.
>
> We would like to clarify that in LMMs, in the second round of conversation, the **input consists of both image and text**, rather than being 'dialogue scenarios with only text.' The Gaslighting Bench constructs image-grounded multi-turn dialogues. In this setting, the model typically answers correctly in the first round; however, the misleading textual cues introduced in the second round cause the model to contradict the visual evidence. Therefore, we attribute this phenomenon to a text-induced visual attention bias.

---

### Note · Authors · 2025-11-20

I have read and agree with the venue's withdrawal policy on behalf of myself and my co-authors.